# Pharmacological Treatment of Patients with Mild to Moderate COVID-19: A Comprehensive Review

**DOI:** 10.3390/ijerph18137212

**Published:** 2021-07-05

**Authors:** Reinaldo B. Bestetti, Rosemary Furlan-Daniel, Vinicius M. R. Silva

**Affiliations:** Department of Medicine, University of Ribeirão Preto, 2201 Costabile Romano, Ribeirão Preto 14096-385, Brazil; rosefurlan@uol.com.br (R.F.-D.); vinicius_mrs@hotmail.com (V.M.R.S.)

**Keywords:** COVID-19, SARS-CoV-2, mild disease, moderate disease

## Abstract

Mild to moderate COVID-19 can be found in about 80% of patients. Although mortality is low, mild to moderate COVID-19 may progress to severe or even critical stages in about one week. This poses a substantial burden on the health care system, and ultimately culminates in death or incapacitation and hospitalization. Therefore, pharmacological treatment is paramount for patients with this condition, especially those with recognized risk factors to disease progression. We conducted a comprehensive review in the medical literature searching for randomized studies carried out in patients with mild to moderate COVID-19. A total of 14 randomized studies were identified, enrolling a total of 6848 patients. Nine studies (64%) were randomized, placebo-controlled trials, whereas five were open-label randomized trials (35%). We observed that Bamlanivimab and nitazoxanide reduced viral load, whereas ivermectin may have shortened time to viral clearance; Interferon Beta-1 reduced time to viral clearance and vitamin D reduced viral load; Favirapir, peginterferon, and levamisole improved clinical symptoms, whereas fluvoxamine halted disease progression; inhaled budesonide reduced the number of hospitalizations and visits to emergency departments; colchicine reduced the number of deaths and hospitalizations. Collectively, therefore, these findings show that treatment of early COVID-19 may be associated with reduced viral load, thus potentially decreasing disease spread in the community. Moreover, treatment of patients with mild to moderate COVID-19 may also be associated with improved clinical symptoms, hospitalization, and disease progression. We suggest that colchicine, inhaled budesonide, and nitazoxanide, along with nonpharmacological measures, based on efficacy and costs, may be used to mitigate the effects of the COVID-19 pandemic in middle-income countries.

## 1. Introduction

COVID-19 illness, which is caused by SARS-CoV-2, has affected about 155 million people throughout the world, killing more than 3.2 million people by May 2021. Although mass vaccination is in progress worldwide, much time will be necessary in order for the overwhelming majority of people to be immunized and the disease to be overcome, especially in developing countries. It is well recognized that nonpharmacological measures—isolation, quarantine, stay-at-home orders, social distancing, gathering prohibition, non-essential business closing, universal mask wearing—are paramount to curb the pandemic. However, evidence-based pharmacological treatment early in the course of the illness, along with nonpharmacological measures, is important to reduce the impact of COVID-19 until vaccine-induced herd immunity is reached.

Mild to moderate COVID-19, defined as the presence of fever, fatigue, dry cough, mild pneumonia, or no pneumonia [1,2], can be found in about 81% of patients with this condition. Up to 14% of patients develop severe COVID-19 illness, characterized by lung infiltrates > 50%, blood saturation < 93%, and dyspnea. More recently, any patient with COVID-19 in need of even low flow supplemental oxygen therapy is considered to have severe COVID-19 illness by the FDA [3]. About 5% of such patients progress to the critical stage (respiratory failure, septic shock, multiple organ dysfunction) [2].

The treatment of patients with mild to moderate COVID-19 is of utmost importance because about 14% of patients can progress to severe COVID-19 in only one week [4,5]; median time to critical COVID-19 appearance may be as early as 8 days from the beginning of symptomatology [1], and the median time to death from the symptom’s onset is 16 days [6]. Therefore, it would be necessary to pharmacologically treat patients with mild to moderate COVID-19 at a high-risk of progression to severe or critical disease. Another reason to treat early COVID-19 is to reduce viral load because viral load not only predicts mortality [7,8] but may also be a risk factor for severe disease; it may also facilitate disease spread [7]. In addition, SARS-CoV-2 peaks at the beginning of the disease, as observed in influenza, thus justifying early treatment of COVID-19 [8].

Several clinical variables have been associated with the progression from mild to moderate COVID-19 illness to severe or critical status, even death, namely those > 75 years old, obesity, chronic kidney disease, diabetes mellitus, hypertension, heart failure, and severe asthma [9,10,11]. Furthermore, laboratory variables—oxygen saturation < 88%, troponin blood levels, C-reactive protein > 200, and D-dimer blood levels > 2500—have also been associated with critical disease [9,10].

Taking into account the facts mentioned earlier, we conducted this review by focusing on the pharmacologic treatment of patients with early COVID-19 (mild to moderate stages) from randomized clinical trials to summarize the available evidence-based support for doctors battling with this scourge worldwide, especially those working in developing countries.

## 2. Methods

We conducted a search in PUBMED using the terms COVID-19 and treatment; we applied the randomized controlled trial filter to refine the search. The objective was to retrieve papers dealing with the treatment of mild to moderate COVID-19, focusing on viral clearance, clinical status, and disease progression following treatment. Therefore, we did not include observational studies in this review. Manuscripts that included patients with either moderate or severe COVID-19 were included in our investigation if the patients with mild to moderate COVID-19 represented more than 50% of total amount.

In addition, papers showing preferentially positive results were included in an attempt to suggest a treatment for doctors dealing with COVID-19. However, when contradictory results were obtained, papers with negative results were included for further discussion. We did not included papers available in preprint servers, which have not undergone peer review. In addition, we also conducted a hand-search of potential references from the previously retrieved papers. Therefore, only papers with a randomized trial performed in patients with mild to moderate COVID-19 illness were potentially included in this investigation.

We retrieved 426 papers that were, at first, read by title and by abstract. When the severity of COVID-19 could not be determined based on titles and abstracts, we read the methods and results section of the papers to make sure that only those that fulfilled the inclusion criteria of our investigation were being enrolled. In so doing, we ruled out 80 studies dealing with severe COVID-19, 194 papers dealing with aspects other than the pharmacological treatment of early COVID-19, 51 papers about structured surveys of a study protocol for a randomized clinical trial in patients with COVID-19, 26 manuscripts related to vaccination, 14 works whose results were inconclusive or only negative, and 8 papers that did not established the severity of the disease. The remaining 37 papers were excluded for the following: not dealing with COVID-19 treatment, a drug from traditional Chinese medicine not available in the Western market, an experimental drug not yet approved by regulatory agencies, or any other reason. A total of 14 papers fulfilled the inclusion criteria and were entered into the study. As the paucity of data concerning a specific treatment for patients with mild to moderate COVID-19 suggests, a formal systematic review or a metanalysis could not be performed.

## 3. Results and Discussion

### 3.1. Micronutrients and Vitamins

#### 3.1.1. Acid Ascorbic and Zinc

Thomas et al. carried out a multicenter, randomized, open-label trial in patients with mild COVID-19 illness. They assigned 58 patients to a zinc group (50 mg/day), 48 patients to an ascorbic acid (8000 mg/day) group, 58 patients to both agents, and 50 patients to standard of care for 10 days. Mean age was 45.2 ± 14.6 years. The primary outcome was the number of days necessary to observe a 50% reduction in symptoms. At the end of study, no difference in the number of days necessary to reach a 50% reduction in symptoms was observed. No side effect that could be ascribed to either zinc or vitamin C was observed [12].

Zinc is believed to stimulate polymorphonuclear cells’ response against viral infection [13], whereas ascorbic acid is an antioxidant that stimulates an immune response [14]. Both may reduce symptom duration and severity in patients with a common cold [14,15]. It is important to emphasize that baseline zinc and ascorbic acid were not measured in the study by Thomas et al. [12]. Therefore, it is not possible to know whether both agents were added to patients with normal or deficient values of these substances. Consequently, we feel that further studies in patients with a zinc and acid ascorbic deficiency, and mild to moderate COVID-19, would be necessary for a consistent conclusion about the role of such substances in the treatment of patients with this condition.

#### 3.1.2. Vitamin D

In a small double-blinded, placebo-controlled clinical trial performed in 40 patients (16 in the intervention group, 24 in the placebo arm), all of them asymptomatic or with mild COVID-19 illness and with a median 25-OH D3 8.8 ng/mL in the intervention group and 9.54 ng/mL in the control group, patients in the intervention group received 60,000 U of cholecalciferol daily for 7 days. Mean age was 48.7 years. The main endpoint was the proportion of patients with a negative SARS-CoV-2 test before day 21, as well as changes in inflammatory markers. Blood levels of 25-OH D3 were significantly higher in the intervention group in comparison to controls; 10 (62%) of 16 patients in the intervention group versus 5 (21%) patients in the control group had negative RNA test for SARS-COV-2 infection (*p* < 0.01). The duration for RNA negativity was similar in both groups. No side effects were observed during the trial, including hypercalcaemia, a common complication of vitamin D excess [16].

Vitamin D protects against viral infections, and patients who received vitamin D supplementation had fewer respiratory infections than those who did not [17]. Patients with COVID-19 with vitamin D deficiency have an increased risk of disease severity and death [18]. Furthermore, the immune-modulatory effect of vitamin D is reached when serum levels are at 25 (OH) [19]. Therefore, vitamin D may be indicated to treat patients with mild to moderate COVID-19 illness when serum levels are low.

### 3.2. Antiviral Drugs

Favipiravir was tested in an open-label, randomized trial that enrolled 72 patients with mild to moderate COVID-19 in the intervention arm and 75 patients in the control arm. Patients were given Favirapir, 1800 mg BID as a loading dose, and 800 mg BID as a maintenance dose for a maximum of 14 days. Mean patient age was 43 ± 11 years. The duration of patients’ symptoms was less than seven days before randomization for patients with mild COVID-19, and <10 days for patients with moderate illness. All patients with moderate symptoms had pneumonia in the chest computed tomography scan and oxygen saturation > 93%. The primary outcome was viral clearance at 28 days. Median time to viral clearance was 5 days in the intervention group compared with 7 days in the control group (*p* = 1.29). However, median time to clinical cure, a prespecified secondary endpoint, was 3 days in the Favirapir group versus 5 days in the control group (*p* = 0.03). Increased blood uric acid and abnormal liver tests occurred in 16% and 7% of patients in the Favirapir group, respectively. Such adverse effects were considered mild to moderate and did not lead to antiviral withdrawal [20].

Hung et al. performed a randomized, open-label study in 86 patients assigned to triple therapy (lopinavir, ritonavir, and interferon beta-1) compared to 41 patients assigned to lopinavir–ritonavir treatment for 14 days; interferon beta-1 was administered subcutaneously in patients with a symptom’s duration < 7 days because of its proinflammatory activity. The efficacy outcome was viral clearance at the study close. Median age was 51 years and 52 years in the triple therapy group and the control group, respectively. Median duration of symptoms before treatment was five days. Median days to negative nasopharyngeal swab was 7 days in the intervention group and 12 days in the control group (*p* = 0.001). In addition, time to complete alleviation of symptoms, a prespecified secondary endpoint, was 4 days in the intervention group and 8 days in the control group (*p* < 0.0001). No statistical difference was observed regarding side-effects between both groups. Serious adverse effects were not observed in the triple therapy group. Nausea, diarrhea, and abnormal liver function tests were similarly observed in the intervention group and in the standard of care group [21].

Remdesivir, a nucleotide prodrug that inhibits viral replication, was studied in an open-label randomized trial enrolling 193 patients who received remdesivir for 10 days, 191 patients who received remdesivir for 5 days, and 200 patients who received standard of care. Remdesivir was given intravenously at a dosage of 200 mg/day on the first day and at a dosage of 100 mg/day thereafter. The primary objective of the study was improvement in clinical status according to a seven-point ordinal scale (1 = death; 2 = invasive mechanical ventilation; 3 = non-invasive ventilation; 4 = low-flow oxygen supplementation; 5 = no oxygen supplementation but in need of medical care; 6 = no oxygen supplementation and no need of medical care; 7 = not hospitalized) by day 11. Median patient age was 57 years. Patients receiving remdesivir for 5 days had a better distribution in the seven-point ordinal scale (fewer patients receiving oxygen supplementation, more patients discharged hospital) in comparison to the other groups. Nausea, hypokalemia, and headache were most frequently observed in the remdesivir groups. However, serious adverse effects that could withdraw the drug studied were more common in the standard of care group than in the remdesivir groups [22].

Favirapir, an oral RNA-dependent RNA polymerase inhibitor, has been used to treat patients with influenza. Therefore, this drug also has the potential to have an inhibitory effect on SARS-CoV-2 infection [23]. The findings obtained by Udvadia et al. [20] hold promise to mitigate symptoms in patients with this condition. Ritonavir and lopinavir have been used to treat SARS-COV and Middle East Respiratory Syndrome (MERS) with success [24]. Interferon Beta has been shown to improve lung pathology and reduce viral load in a marmoset model of MERS-COV infection [25]. This was the rationale for the study by Hung et al. [21], which was associated with symptoms’ mitigation, reduced duration of viral shedding, and hospitalization in patients with mild to moderate COVID-19. Remdesivir inhibits viral RNA-dependent RNA polymerase through a metabolite, thus interfering with viral replication. Remdesivir has been useful in time to recovery in patients with severe COVID-19. In a study by Spinner et al. [22], remdesivir was of value to improve clinical status. Collectively, the studies mentioned earlier may be useful in the treatment of early COVID-19.

### 3.3. Anti-Inflammatory Drugs

#### 3.3.1. Colchicine

A large study has recently been reported regarding the efficacy of colchicine in patients with mild to moderate COVID-19. In a COLCORONA trial, a total of 4488 patients with this condition were included in the investigation; 2235 patients were randomly assigned to the colchicine group and 2253 patients were assigned to the placebo group before confirmation of the diagnosis of COVID-19, which was subsequently confirmed in 93% of patients. Colchicine was given at the dosage of 0.5 mg twice a day for the first three days, and once a day for the remaining 27 days of the trial. Mean symptoms duration before randomization was 5.4 ± 4.4 days. Median age was 53 years in the colchicine group and 54 years in the placebo group. A sensitivity analysis performed in patients with confirmed COVID-19 illness showed that the primary efficacy endpoint occurred in 96 (5%) of 2075 patients in the colchicine group, and in 126 (6%) of 2084 patients in the placebo group. Such a difference in the composite endpoint was driven by a reduction of 25% in hospitalization for COVID-19 (hazard ratio = 0.75; 95% confidence interval 0.57–0.99; *p* = 0.04). Serious adverse effects occurred more frequently in the placebo group than in COVID-19 patients, including severe pneumonia. Nonetheless, diarrhea was observed more frequently in the intervention group than in the placebo group, but this did not preclude patients from receiving the medication [26].

Colchicine inhibits the inflammasome-induced proinflammatory cytokines release that is secondary to viral infection [27], as well as neutrophil chemotaxis and platelet aggregation [28]. This is the rationale for its use in the treatment of several inflammatory diseases, mainly gout and acute pericarditis [28]. Therefore, the study by Tardiff et al. [26] clearly shows that colchicine can be given to patients in the outpatient setting, even before the confirmation of the illness, to halt clinical deterioration to the point of needing hospitalization; there are also very few serious adverse effects.

#### 3.3.2. Fluvoxamine

Fluvoxamine was studied in patients with mild COVID-19 in a randomized, double-blinded, placebo-controlled clinical trial enrolling 80 outpatients in the intervention group (fluvoxamine, 100 mg three times a day) and 72 outpatients in the placebo arm. Median patient age was 46 years; comorbidities were frequently included and 54% of the patients were obese. Median duration of symptoms was 4 days. The main efficacy endpoint was clinical deterioration within 15 days of randomization, defined by appearance of shortness of breath or hospitalization and/or oxygen saturation < 92% or need of oxygen therapy. Clinical deterioration was found in 0% of patients in the fluvoxamine group and 6 (9%) of patients in the placebo group (*p* = 0.009). Headache, nausea and vomiting, and muscle aches were the side effects most frequently found in both groups without statistical significance. No difference was observed in the percentage of side effects between both groups (15.1% in intervention arm x 15.2% in the placebo arm) [29].

Fluvoxamine, a selective serotonin reuptake inhibitor, may have a role in the treatment of patients with mild to moderate COVID-19 illness due to its anti-inflammatory, antiviral, and inhibitory platelet activation properties. Fluvoxamine acts through strong stimulation of the alpha-1 receptor in the endoplasmic reticulum, thus reducing the inflammatory response to infection by regulating cytokine production [30]. Therefore, fluvoxamine has a potential for the treatment of early COVID-19, but larger studies will be necessary to confirm this beneficial effect.

### 3.4. Antiparasitic Drugs: Nitazoxanide, Ivermectin, and Levamisole

The efficacy of nitazoxanide was evaluated in a multicenter, double-blinded, randomized, placebo-controlled trial enrolling 392 patients with mild COVID-19 (198 assigned to placebo group and 194 to nitazoxanide group) diagnosed by a RT-PCR test for SARS-CoV-2. Median time from initial symptoms to the first drug use was 5 (4–5) days. About 500 mg of nitazoxanide was given thrice to patients in the intervention group for five days. About 94% of patients were aged 18 to 50 years. Primary outcome was symptoms resolution (fever, dry cough, and fatigue), and the secondary outcome was viral load. Following 5 days of treatment, no difference was found between the intervention group and the placebo group regarding symptom resolution, which occurred in about three quarters of patients. However, median RT-PCR viral load was 3.63 (0–5.03) log_10_ copies/mL in the nitazoxanide arm and 4.13 (2.88–5.33) log_10_ copies/mL in the placebo arm (*p* = 0.006) [31]. Diarrhea, headache, and nausea were the side effects similarly observed in both groups; the proportion of patients who were withdrawn from the study because of adverse effects was also similar in both groups (3.1% in the treatment arm x 0.5% in the placebo arm).

The impact of ivermectin on viral clearance has also been studied. Ahmed et al. [32] randomized 72 patients to oral ivermectin (*n* = 24, 12 mg/day for 5 days), oral ivermectin + doxycycline (*n* = 24), and placebo (*n* = 24). Despite hospitalization, all patients had mild to moderate disease. Mean age was 42 years. Mean duration of the disease before enrolment was 3.8 days. The efficacy outcome was time to viral clearance, as well as remission of fever and cough within 7 days. Mean duration to viral clearance was 9.7 days in the ivermectin group, 11.5 days in the ivermectin + doxycycline group, and 12.7 days in the placebo arm (*p* = 0.005). Kaplan–Meier analysis showed that the proportion of patients at risk of viral infection was reduced in the ivermectin group (*p* = 0.02). In addition, C-reactive protein was also reduced in the ivermectin group in comparison with the other groups, thus suggesting that ivermectin may reduce disease severity. No adverse effect was observed in both groups [32].

In another study, 476 patients with mild disease were enrolled in a randomized, double-blinded, placebo control trial; the intervention group received ivermectin, 300 ug/kg/body weight, for 5 days. The primary outcome was time to resolution of symptoms in a 21-day follow-up. Median age was 37 years. Median time for receiving ivermectin was 5 days; 61% of patients had no limitation of activities and were not hospitalized; 37% patients received oxygen therapy at home. By day 21 of follow-up, no difference was observed between both groups. Headache, dizziness, diarrhea, and nausea were the side effects more frequently observed in both groups, but no statistical difference was detected between groups [33].

Nitazoxanide inhibits SARS-C0V-2 replication in Vero CCL81 cells [34]. In a study by Rocco et al., nitazoxanide was able to reduce viral load and increase the number of patients with negative results by day 5. Therefore, considering the trial design, the sample size, and the hard endpoint, nitazoxanide appears to be a promising drug for the treatment of early COVID-19.

The use of ivermectin in early COVID-19 might be important from a clinical and epidemiological viewpoint because viral load is associated with mortality in patients with COVID-19 [35], as well as being inexpensive and safe. Ahmed and colleagues showed a decrease in viral clearance, but the sample size was too low to allow for a firm conclusion. The apparent contradiction between both studies related to ivermectin may have occurred because ivermectin was given too late in the study by Medina et al. [33], since the drug should be given at the beginning of symptoms’ onset [32]. Therefore, further studies are necessary to clarify the role of ivermectin in the treatment of early COVID-19.

Levamisole was evaluated in a randomized, placebo-controlled trial (25 patients in each arm of the investigation) in patients with mild to moderate COVID-19, defined as those with oxygen saturation > 94% and abnormalities typical of COVID-19 on a chest computed tomography scan. Levamisole was given at a dose of 50 mg by mouth, three times a day for three days. All patients were treated on an ambulatory basis. Median age was 37 years. At baseline, mean oxygen saturation was 96%, and 88% of patients had positive CT scan. On day 14 of levamisole treatment, 9 (36%) patients in the placebo arm and 1 (4%) patient in the intervention group had a cough (*p* = 0.005); on day 7, 12 (48%) patients in the placebo group and 4 (16%) patients in the intervention group had dyspnea (*p* = 0.01). However, the proportion of patients with dyspnea was higher in the intervention group than in the placebo group on day 14 of the study. No side effects were observed in either group [36].

Levamisole, a synthetic molecule, improves cellular and humoral immunity by regulating the release of proinflammatory cytokines, normalizing the CD4+/CD8+ cell ratio, and elevating IGA and IGM serum levels [36]. This is the mechanism by which levamisole was associated with the favorable results in the study by Firozabad et al. [36]. However, since the sample size was small and the endpoint was not challenging, more studies are necessary to evaluate the role of levamisole in the treatment of patients with mild to moderate COVID-19.

### 3.5. Neutralizing Antibodies

In a BLAZE study, a multicenter, randomized, placebo-controlled trial, a total of 577 patients with mild to moderate COVID-19 were enrolled. Of these, 449 (78%) patients had mild disease, whereas the remaining patients had moderate COVID-19. Median age of patients was 44 years. Median time from symptom onset to first drug use was 4 days. Bamlanivimab at a dosage of 700 mg/day was compared with Bamlanivimab at a dosage of 2800 mg; Bamlanivimab at a dosage of 7000 mg/daily and Bamlanivimab at a dosage of 2800 mg/day were associated with etesivimab (2800 mg/day) and placebo. The main objective was to determine viral load at day 11 of the study. In comparison with the placebo arm, no difference in the viral load was observed among the several groups of Bamlanivimab monotherapy. Nonetheless, Bamlanivimab associated with etesivimab markedly reduced viral load. Only one patient in the combination group developed a urinary tract infection, which was unrelated to the drugs themselves. Nausea and diarrhea were detected in less than 5% of both groups [37].

Monoclonal neutralizing antibody has been shown to be effective in the treatment of viral illnesses [37]. Bamlanivimab and etesivimab are anti-spike neutralizing monoclonal antibodies manufactured from the serum of patients who have recovered from SARS-CoV-2 infection. Gottlieb and co-workers clearly showed in this well-designed trial that this combination of neutralizing antibodies has a venue for the treatment of outpatients with mild to moderate COVID-19 because of its efficacy and safety in reducing viral load. This is important, as mentioned earlier, as viral load predicts mortality [8] and morbidity [7], as well as contributing to SARS-CoV-2 spread in the community [8].

REGN-COV-2, a cocktail containing two neutralizing antibodies against SARS-CoV-2, was tested in patients with moderate COVID-19. Weinreich and colleagues reported the results of a randomized placebo-controlled trial enrolling 275 patients in total; 92 patients were assigned to receive REGN-COV2, 2.4 g diluted in 250 mL of saline solution and given intravenously for 1 h, and 90 patients were assigned to receive REGN-COV2, 8 g, as previously mentioned. A further 93 patients were assigned to the placebo group. Median patient age was 44 years. Compared with the placebo group, viral load was significantly lower in both groups of patients receiving daily REGN-COV-2. In addition, the percentage of patients in need of medical assistance was lower in the REGN-COV2 groups in comparison to patients assigned to the placebo group. No difference was observed regarding the proportion of side effects in REGN-COV2 groups compared with the placebo group [38].

REGN-2 is another combination of neutralizing antibodies that hold promises for the treatment of patients with mild to moderate COVID-19. The study by Weinreich et al. [38] highlights that REGN-2 also improves clinical status, as well as reducing viral load. Thus, this class of drug may be considered for the treatment of early COVID-19.

### 3.6. Interferons

Feld et al. studied the potential usefulness of peginterferon in patients with mild to moderate COVID-19 [38]. They enrolled 60 patients with this condition; 30 patients were randomly assigned to the peginterferon group (one single 180 μg subcutaneous injection) and 30 patients were assigned to the placebo group. The main efficacy outcome was a negative SARS-CoV-2 mid-turbinate swab by day 7, whilst the main safety endpoint was serious adverse effects by day 14. Median patient age was 46 years. Compared to the placebo group, those patients with a baseline viral load of 10^6^ copies per mL in the intervention group had a marked increase in undetectable virus on day 7 (79% in the intervention group x 38% in the placebo group; *p* = 0.01). No difference was observed in the proportion of adverse effects between both groups, except for mild and transient aminotransferase increase in the intervention group, which is the most common complication of peginterferon administration [39].

Interferons are produced by the human organism in response to viral infections. Interferon lambda, a type III interferon, stimulates antiviral activity without producing a cytokine storm (particularly in the lungs), and is better tolerated than interferons type I and II [40]. Peginterferon has been used as treatment in patients with viral hepatitis [41].

This study suggests that peginterferon may be an alternative for treating patients with mild to moderate COVID-19 to reduce viral load. However, because of the small sample size, further larger studies are needed to determine the usefulness of interferons in the treatment of early COVID-19.

### 3.7. Glucocorticoids

Inhaled budesonide was used in an open-label study in the treatment of COVID-19. Ramakrishnam et al. randomized 70 patients to the budesonide arm (2 puffs = 800 μg twice a day for a median duration of 7 days) and 69 patients to usual care. Median symptom duration before randomization was 3 days. Primary outcome was urgent care visits to the emergency department or hospitalization. Patients’ follow-up was 28 days. Median patient age was 45 years. The primary outcome occurred in 11 (15%) patients in the usual care arm and in 2 (3%) patients in the budesonide arm (HR = 0.23; 95% CI 0.03 to 0.21, *p* = 0.009). The number needed to be treated with budesonide to halt disease progression was 8 [5].

Inhaled glucocorticoids have been shown to reduce SARS-CoV-2 replication in airways epithelial cells in vitro [42]. They have been extensively used in the treatment of patients with chronic obstructive pulmonary disease and asthma to reduce exacerbation periods, which are believed to be caused by viral infections [5]. The studied mentioned above shows that inhaled budesonide is safe and efficacious in the setting of SARS-COV-2 infection. For this reason, and other than being relatively inexpensive, inhaled budesonide may have a role in the treatment of early COVID-19 illness.

### 3.8. Nasal Irrigation with Hypertonic Saline

A small randomized, open-label study was carried out in 45 patients with mild COVID-19; 17 received no intervention, 14 were treated with nasal irrigation with 250 mL of hypertonic saline twice daily, and 14 in the group received hypertonic saline (as described earlier) with 1 mL of surfactant. Mean age was not reported. Median time of symptoms’ duration was two days. It is important to emphasize that patients performed nasal irrigation while isolated in a bathroom to avoid viral dissemination. Median time of nasal congestion resolution was 14 days in the non-intervention group, seven days in the group treated with hypertonic saline and surfactant, and five days in the group treated with hypertonic saline only (*p* = 0.04). In addition, time to headache resolution was 12 days in the non-intervention group, five days in the hypertonic saline with surfactant, and three days in the group that received hypertonic saline alone (*p* = 0.02) [42].

Therefore, this study has promise regarding an inexpensive treatment without important side effects. However, further larger studies will clearly be necessary to establish the role of hypertonic saline in the treatment of mild COVID-19.

Table 1 summarizes the studies analyzed in this review.

## 4. Conclusions

Based on the manuscripts reviewed and mentioned earlier, it should be clear that there is evidence-based medicine support for the pharmacological treatment of early COVID-19. This support relies on the impact of some drugs on the shortening time for SARS-CoV-2 clearance, which is important not only in terms of reducing mortality but also for decreasing the spread of the disease. In addition, some drugs may be important to decrease disease progression from mild to moderate to severe stages of COVID-19. Taking into account the large sample size, the placebo-controlled trial, the hard endpoint, low cost, and safety, colchicine seems to be a good indicator for early COVID-19 reductions in deaths and hospitalizations, especially in developing countries. At the same time, Bamlanivimab + etesivimab may be attractive for reducing viral load, but the cost-effectivity of this therapeutic remains to be determined. The same can be said regarding the use of remdesivir to improve clinical status. Budesonide can be used to reduce the number of visits to emergency departments and hospitalizations, while nitazoxanide is suitable for reducing viral load, and deserves consideration for the treatment of patients with COVID-19 early on in the course of the illness.

Early COVID-19 treatment may be especially effective in developing countries where mass vaccination is in progress, but only on a small scale. In such countries, non-pharmacological measures to curb the pandemic are difficult to deploy because of social inequalities and illiteracy, and these methods might be associated with the appearance of dangerous SARS-CoV-2 variants [44], themselves a potential worldwide threat. Furthermore, in such countries, epidemiological surveys suggest that early COVID-19 treatment may be associated with decreased mortality [45]. For the time being, taking into account the potential cost-effectiveness and the initial support of evidence-based medicine, we suggest that nitazoxanide, budesonide, and preferentially colchicine should be used in the treatment of early COVID-19 in an attempt to mitigate the scourge that this disease has been causing around the world. However, we concede that further studies will be necessary to improve the robustness of evidence-based medicine used to support the treatment of early COVID-19.

## Figures and Tables

**Table 1 ijerph-18-07212-t001:** Summary of the randomized studies performed in patients with mild to moderate COVID-19 showing beneficial effects in patients with this condition.

Authors	Number of Patients	Type of Study	Used Drug	Outcome According to Authors of the Papers Examined
Gottlieb et al. [37]	577	Placebo-controlled	Bamlanivimab + etesivimab	Reduced viral load
Ramakrishnam et al. [5]	139	Open-label	Inhaled Budesonide	Reduced hospitalization and number of visits to emergency department
Rastogi et al. [16]	40	Placebo-controlled	Vitamin D	Reduced viral load
Udwadia et al. [20]	147	Open-label	Favirapir	Shortened time to cure
Hung et al. [21]	127	Open-label	Interferon beta-1 + L + R	Shortened time to viral clearance
Tardiff et al. [26]	4488	Placebo-controlled	Colchicine	Reduced the composite endpoint of death or hospitalization
Lenze et al. [29]	152	Placebo-controlled	Fluvoxamine	Reduced clinical deterioration
Rocco et al. [31]	392	Placebo-controlled	Nitazoxamide	Reduced viral load
Ahmed et al. [32]	72	Placebo-controlled	Ivermectine	May have shortened time to viral clearance
Firozabad et al. [36]	50	Placebo-controlled	Levamisole	May reduce clinical symptoms
Kimura et al. [43]	45	Open-label	Hypertonic saline	Reduced clinical symptoms
Spinner et al. [22]	584	Open-label	Remdesivir	Improved clinical status
Weinreich et al. [38]	275	Placebo-controlled	REGN-COV2	Decreased viral load and improved clinical status
Feld et al. [39]	60	Placebo-controlled	Peginterferon lambda	Decreased viral load

L = Lopinavir; R = Ritonavir.

## Data Availability

Not applicable.

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
