# Peer review of "Pharmacological Treatment of Patients with Mild to Moderate COVID-19: A Comprehensive Review"

_ijerph, 2021, doi:10.3390/ijerph18137212_

Round 1

Reviewer 1 Report

This manuscript reviewed the effects of pharmacological treatment of COVID-19 in the medical literature searching for randomized studies updated to May of 2021. Mild to moderate COVID-19 can be found in about 80% of patients. Thus, early treatments of COVID-19 patients are important in the current stage. The manuscript is recommended for publication. Some suggestion is listed for the consideration of revision.

  1. Bamlanivimab and nitazoxanide reduced viral load; ivermectin and Interferon Beta-1 reduced time to viral clearance; Favirapir reduced time of viral shedding; levamisole improved clinical symptoms; ….. The authors are suggested to come up with some discussion to explain why these drugs led to the medicinal effects.
  2. For a cautious information, the authors may provide the side effects of those drugs observed in the randomized studies.

Author Response

This manuscript reviewed the effects of pharmacological treatment of COVID-19 in the medical literature searching for randomized studies updated to May of 2021. Mild to moderate COVID-19 can be found in about 80% of patients. Thus, early treatments of COVID-19 patients are important in the current stage. The manuscript is recommended for publication. Some suggestion is listed for the consideration of revision.

Thanks for the comments. We thank referee’s suggestions, which have improved the quality of our paper. 

Bamlanivimab and nitazoxanide reduced viral load; ivermectin and Interferon Beta-1 reduced time to viral clearance; Favirapir reduced time of viral shedding; levamisole improved clinical symptoms. The authors are suggested to come up with some discussion to explain why these drugs led to the medicinal effects.

Concordamos com a sugestão do árbitro. Uma explicação para o efeito de cada medicamento é fornecida ao longo do manuscrito alterado (por favor, veja a nova seção Resultados e Discussão agora fornecida na versão revisada do manuscrito; linhas: 97 a 104; 116-120; 146 a 168; 186 a 191; 203-208; 240 a 250; 261 a 265; 279 a 285; 296 a 299; 312 a 318; 328 a 333). 

Para uma informação cautelosa, os autores podem fornecer os efeitos colaterais dessas drogas observados nos estudos randomizados.

Feito. Os efeitos colaterais observados nos estudos examinados, bem como seu impacto na saúde dos pacientes, foram incluídos em toda a versão revisada do manuscrito (linhas: 95 a 96; 114 a 115; 132 a 134; 144 a 145; 155 a 157; 182 a 1851; 199 a 202; 220 a 222; 231 a 232; 259 a 260; 276 a 278; 293 a 295; 308 a 311.

Reviewer 2 Report

Dear authors:

After reading the manuscript entitled “Effects of early pharmacological treatment of COVID-19 on disease spread, morbidity, and hospitalization: A comprehensive review”, I consider that the topic of the paper is interesting and relevant. Knowing better the possible pharmacological treatments against COVID-19 is essential to overcome this pandemic that is putting the world into a big threat.

We must not forget that pharmacological treatment must go in parallel with nonpharmacological measures

Given that 80% of the patients have mild to moderate COVID-19 and that up to 14% of patients develop severe COVID-19 illness, it seems interesting to know the possible pharmacological treatments that can improve clinical symptoms, reduce viral load or shorten the virus elimination time. It is also important to take into account the correlation between the viral load and the Morbidity and Mortality.

Major comments:

As it is a review article, although the authors advance in the Search Strategy section that there are few articles. I suggest the inclusion of a greater number of articles in order to reach more solid conclusions. On many occasions, only one study is cited to support the efficacy / non-efficacy or use of a particular treatment. I consider interesting to carry out a deeper review in order to include more articles, giving more solidity to the paper.

On the other hand, and as a personal recommendation with the aim of turning this review article into an excellent and complete review, I encourage authors to enlarge the inclusion criteria, covering not only the disease in its mild-moderate versions but also in the severe one. They also can compare treatments and results.

In order to improve the understanding of the article, I recommend the authors to carry out a reorganization of the content in the following sections: Introduction, Material and methods, Results and Discussion, Conclusion. The Result and Discussion section should include the different types of treatments and a brief discussion of each one of these treatments. Another possibility is to divide Results and Discussion in two sections.

Inconsistencies are detected between the results presented in the text and the statements that are given in Table 1. This can lead to mistakes regarding the possible treatments. In addition, it should be emphasized that there are many studies with a small study population size, which does not allow the results obtained to be extracted or extrapolated. This fact should be stated due to, as indicated by the authors of the experimental studies, more studies should be carried out to reach conclusive results.

The title does not adequately reflect the content of the paper. A evaluation is proposed.

Minor comments:

According to the abstract, the review focuses on 11 articles that meet the inclusion criteria, however when classifying them I only find 10 of them. There is also a mismatch between the number of patients included in studies (1323) according to the abstract and the total number of patients that appear in Table 1. These data should be clarified and corrected.

In line 33, and with respect to non-pharmacological measures, authors should indicate which are these measures in order to give a more complete overview of the introduction.

In line 41 of the manuscript, FDA is referenced. A bibliographic reference in which the information provided can be checked is missing.

In line 92 there is a mistake regarding the data of the original study: “… vs 5 (21%) patients in the control group had…”

Line 94, change RNE for RNA.

In line 157: the study carried out by Cavalcanti (2020) appears referenced. However, I cannot find this study in the bibliographic references.

Line 168, explain briefly what the 7-level ordinal scale consists of and its relationship with COVID-19.

Line 184 and 185, there are missing units in the values indicated.

Line 196, what dose CRP mean?

Line 253: the text that explaine Table 1 should be inserted in a separate paragraph from the previous text that talks about glucocorticoids.

Regarding the writing and the expression of the manuscript, I would like to propose changing certain expressions that are repeated throughout the text and that can result in a somewhat tedious repetition, such as "The primary outcome was ..." in line 77, 89, 106, 115, 167 …

When synthesizing the studies and extracting the most relevant information for this current review, some data is sometimes missing, such as the population size in the Acid Ascorbic and Zinc sections or the age of the patients in the study concerning the use of ivermectin.

In addition, there are studies included in the review that do not seem to meet the previously established inclusion criteria by the authors. As for example, the study in which the treatment is evaluated with colchicine, where the 97% of the patients have a severe disease or others studies related to chloroquine and Hydroxychloroquine .

In Table 1, in the outcome column, the conclusions should be expressed with more caution since there are cases (for example in ivermectin or levamisole) in which it cannot be firmly stated that treatment reduces viral load or improves clinical symptoms.

Author Response

After reading the manuscript entitled “Effects of early pharmacological treatment of COVID-19 on disease spread, morbidity, and hospitalization: A comprehensive review”, I consider that the topic of the paper is interesting and relevant. Knowing better the possible pharmacological treatments against COVID-19 is essential to overcome this pandemic that is putting the world into a big threat. We must not forget that pharmacological treatment must go in parallel with nonpharmacological measures. Given that 80% of the patients have mild to moderate COVID-19 and that up to 14% of patients develop severe COVID-19 illness, it seems interesting to know the possible pharmacological treatments that can improve clinical symptoms, reduce viral load or shorten the virus elimination time. It is also important to take into account the correlation between the viral load and the Morbidity and Mortality.

We are grateful to the referee’s criticism. All changes in the amended manuscript have been marked in red in the revised text.

Major comments:

As it is a review article, although the authors advance in the Search Strategy section that there are few articles. I suggest the inclusion of a greater number of articles in order to reach more solid conclusions. On many occasions, only one study is cited to support the efficacy / non-efficacy or use of a particular treatment.  I consider interesting to carry out a deeper review in order to include more articles, giving more solidity to the paper.

Thanks for the suggestion. We have used a different approach in the search strategy for articles related to pharmacologically treatment of early COVID-19. We enter PUBMED using the terms COVID-19 and treatment, filtered by the words Randomized Clinical Trials. By doing that, we retrieved 426 papers, much more than we have retrieved before with another search strategy. We read the papers title and abstract to rule out patients with severe COVID-19 and include patients with only mild to moderate COVID-19; when this did not suffice, we read the methods and results sections of each paper to properly identify manuscripts dealing with pharmacologically treatment of early COVID-19. After that, all selected papers were read in full. By doing that, we have been able to find five additional papers on pharmacologically treatment for early COVID-19, and we removed two papers on chloroquine and hydroxychloroquine because new identified papers have showed no benefit of these drugs to treat patients with mild to moderate COVID-19. Overall, 6848 patients from a total of 14 studies have been incorporated into the amended manuscript. The additional papers have been marked in red in the reference list (please, see lines 13 to 19).

On the other hand, and as a personal recommendation with the aim of turning this review article into an excellent and complete review, I encourage authors to enlarge the inclusion criteria, covering not only the disease in its mild-moderate versions but also in the severe one. They also can compare treatments and results.

We agree and thank the referee suggestion to enlarge the manuscript by including papers with severe COVID-19. However, as highlighted in the Methods section of the amended manuscript, we have excluded 82 papers dealing with severe COVID-19.  Therefore, the number and the excellence of papers dealing with a specific treatment of severe COVID-19 might enlarge too much the current paper. In addition, we feel that the focus of our work – the pharmacological treatment of early COVID-19 - might be lost. In addition, we have identified additional more five additional papers, which have been included in the revised version of the manuscript. In our view, the 14 papers dealing with pharmacologically treated-patients with mild to moderate COVID-19, which have included a total of 6848 patients in the final manuscript version, will increase the robustness of our work.

In order to improve the understanding of the article, I recommend the authors to carry out a reorganization of the content in the following sections: Introduction, Material and methods, Results and Discussion, Conclusion. The Result and Discussion section should include the different types of treatments and a brief discussion of each one of these treatments. Another possibility is to divide Results and Discussion in two sections.

We agree with reviewer suggestion. In the revised version of the manuscript, the sections Introduction, Methods, Results and Discussion, and Conclusions have now been included. Furthermore, a brief comment on each treatment is given throughout the amended manuscript.

(Please, see lines 64, 88, and 354).

Inconsistencies are detected between the results presented in the text and the statements that are given in Table 1. This can lead to mistakes regarding the possible treatments. In addition, it should be emphasized that there are many studies with a small study population size, which does not allow the results obtained to be extracted or extrapolated. This fact should be stated due to, as indicated by the authors of the experimental studies, more studies should be carried out to reach conclusive results.

We agree with referee’s criticism. The inconsistencies have been removed from the text of the amended manuscript (see, please, the Abstract section and Table 1). In the first and in the second paragraphs of the Abstract section of the revised version of the manuscript, we have emphasized that further studies will be necessary to draw firm conclusions about the treatment of early COVID-19 (please, lines 13 to 19). Furthermore, we have emphasized the papers with large sample size, focusing on hard endpoints, including a placebo-controlled design, may be useful to treat patients with this condition (and 23 to 26).  

The title does not adequately reflect the content of the paper. An evaluation is proposed.

We agree with the referee suggestion. The title has been changed accordingly. (Line 2)

Minor comments:

According to the abstract, the review focuses on 11 articles that meet the inclusion criteria, however when classifying them I only find 10 of them. There is also a mismatch between the number of patients included in studies (1323) according to the abstract and the total number of patients that appear in Table 1. These data should be clarified and corrected.

The referee is right. It is now clearly emphasized in the amended manuscript that the review focus on 14 papers, enrolling 6848 patients. Such inconsistency does not exist anymore. (see lines 12 to 14).

In line 33, and with respect to non-pharmacological measures, authors should indicate which are these measures in order to give a more complete overview of the introduction.

Right. We have included a new statement in the amended manuscript (please, see the Introduction section, first paragraph, lines 35 to 40) indicating such non-pharmacological measures.

In line 41 of the manuscript, FDA is referenced. A bibliographic reference in which the information provided can be checked is missing.

  1. This reference has now included in the reference list of the revised version of the manuscript. (please, see reference number 3 of the amended manuscript, lines 394 to 396).

In line 92 there is a mistake regarding the data of the original study: “… vs 5 (21%) patients in the control group had…”

We are grateful to the referee’s correction. The mistake has been corrected in the amended manuscript (please, see the first paragraph of Vitamin D section, line 112)

Line 94, change RNE for RNA.

This has been corrected. Please, see the first paragraph of Vitamin D section (line 113).

In line 157: the study carried out by Cavalcanti (2020) appears referenced. However, I cannot find this study in the bibliographic references.

The referee is wright. As explained in the text of the amended manuscript, the section concerning hydroxychloroquine has been removed from the text because we found new references showing a lack of effects of such drugs in patients with mild to moderate COVID-19. Consequently, the work by Cavalcanti et. al has been removed as well. However, we have used these manuscripts on chloroquine and hydroxychloroquine as part of the new discussion in the corrected text (please, see the second paragraph of colchicine section).

Line 168, explain briefly what the 7-level ordinal scale consists of and its relationship with COVID-19.

Done. The 7-level ordinal scale created by the WHO has been explained in detail (please, see the third paragraph of the antiviral drugs, remdesivir section, lines 150 to 152).

Line 184 and 185, there are missing units in the values indicated.

The units related to the viral load has been included in the amended manuscript (please, see the first paragraph of the nitazoxanide section, lines 218 to 219)

Line 196, what dose CRP mean?

CRP means C Reactive Protein. This has now been detailed in the revised version of the manuscript (see line 230).

Line 253: the text that explained Table 1 should be inserted in a separate paragraph from the previous text that talks about glucocorticoids.

Done. Table 1 has been inserted in a separate paragraph in the new section Nasal irrigation of the revised version of the manuscript (please, see the last paragraph of Results and Discussion section of the revised manuscript, line 348).

Regarding the writing and the expression of the manuscript, I would like to propose changing certain expressions that are repeated throughout the text and that can result in a somewhat tedious repetition, such as "The primary outcome was ..." in line 77, 89, 106, 115, 167 …

Ok. We have changed the expressions according to referee’s suggestion throughout the manuscript. By doing that, we believe that there is no more tediously expression repetition.

When synthesizing the studies and extracting the most relevant information for this current review, some data is sometimes missing, such as the population size in the Acid Ascorbic and Zinc sections or the age of the patients in the study concerning the use of ivermectin.

The referee is correct. Mean age is provided in each summarized study. The sample size concerning Acid Ascorbic and Zinc section has been given in the amended manuscript (lines 93, 109, 126, 139, 153, 177 to 178, 195, 215, 225, 235, 270, 290, 306, 325, 338).

In addition, there are studies included in the review that do not seem to meet the previously established inclusion criteria by the authors. As for example, the study in which the treatment is evaluated with colchicine, where the 97% of the patients have a severe disease or others studies related to chloroquine and Hydroxychloroquine.

We agree with the referee criticism. For this reason, the study by Deftereos et al, in which 66% of patients had severe, and not mild to moderate COVID-19, were removed from the study. We also removed the study by Lopes et al because the vast majority of patients had severe COVID-19. However, we included another study performed in patients with early COVID-19, which form the basis for colchicine recommendation to patients with this condition (please, see Tardiff et.al in the reference list of the revised manuscript). The question about the chloroquine and hydroxychloroquine is very problematic. We have included the text in the original manuscript because we had found only Cavalcanlti paper dealing with moderate COVID-19. As we had pointed out in the original manuscript, Cavalcanti et al. have studied patients with severe -COVID-19. Therefore, we had tried to show that hydroxychloroquine had not been studied in patients with mild to moderate COVID-19. However, we have found some papers showing that hydroxychloroquine has no beneficial effect in patients with early COVID-19. Consequently, we decided to remove that paragraph in the amended manuscript because it does not make sense anymore. Nonetheless, the references Lopes et al. and Deftereos et al. have been used for a brief discussion on colchicine characteristics in the colchicine section in the amended manuscript.

In Table 1, in the outcome column, the conclusions should be expressed with more caution since there are cases (for example in ivermectin or levamisole) in which it cannot be firmly stated that treatment reduces viral load or improves clinical symptoms.

We understand the referee observation. However, the outcome we have included in the Table is the conclusion of the authors of the papers used in the review. To make clear in the amended manuscript, we have changed the last column “outcome” to “outcome according to authors of the papers examined”. In addition, we have included the word “may” before the words “reduce clinical symptoms”, according to Firozabad et al. conclusion. Furthermore, we have replaced the terms ‘reduced viral load” in Table 1 of the original manuscript by the terms “may have shortened time to viral clearance” to keep in line with Ahmed et. conclusion. Please, see Table I of the revised version of the manuscript, lines 350 to 352 .

Round 2

Reviewer 2 Report

Dear authors:

After second revision of the manuscript, I appreciate the clarifications of the authors and the improvements made. Thus, the paper is recommended for publication.